# Photography as the Sole Means of Proof: Medical Liability in Dermatology

**DOI:** 10.3390/diagnostics13061033

**Published:** 2023-03-08

**Authors:** Maricla Marrone, Cristina Caterino, Gianluca Musci, Gerardo Cazzato, Giuseppe Ingravallo, Carmelo Lupo, Nadia Casatta, Alessandra Stellacci, Andrea Armenio

**Affiliations:** 1Section of Legal Medicine, Department of Interdisciplinary Medicine (DIM), University of Bari “Aldo Moro”, 70124 Bari, Italy; 2Section of Molecular Pathology, Department of Precision and Regenerative Medicine and Ionian Area (DiMePRe-J), University of Bari “Aldo Moro”, 70124 Bari, Italy; 3Innovation Department, Diapath S.p.A., Via Savoldini n.71, 24057 Martinengo, Italy; 4Plastic Surgery Department, IRCCS Istituto Tumori “Giovanni Paolo II”, V.le O. Flacco, 65, 70124 Bari, Italy

**Keywords:** legal medicine, malignant melanoma, defensive medicine, histopathology, dermatology

## Abstract

Malignant melanoma is a cutaneous malignancy resulting from the uncontrolled proliferation of melanocytes and poses a challenge diagnostically because neoplastic lesions can mimic benign lesions, which are much more common in the population. Doctors, when they suspect the presence of melanoma, arrange for its removal and the performance of a histological examination to ascertain its diagnosis; in cases where the dermatoscopic examination is indicative of benignity, however, after the lesion is removed, histological examination is not always performed, a very dangerous occurrence and a harbinger of further medico-legal problems. The authors present a court litigation case of an “alleged” failure to diagnose malignant melanoma in a patient who died of brain metastases from melanoma in the absence of a certain location of the primary tumor: the physician who had removed a benign lesion a few months earlier was sued, and only thanks to the presence of photographic documentation was the health care provider able to prove his extraneousness. The aim of this paper is to formulate a proposal for a dermatological protocol to be followed in cases of excisions of benign skin lesions with a twofold purpose: on the one hand, to be able to prove, in a judicial context, the right action on the part of the sanitarians; on the other hand, to avoid the rise of so-called “defensive medicine”.

**Figure 1 diagnostics-13-01033-f001:**
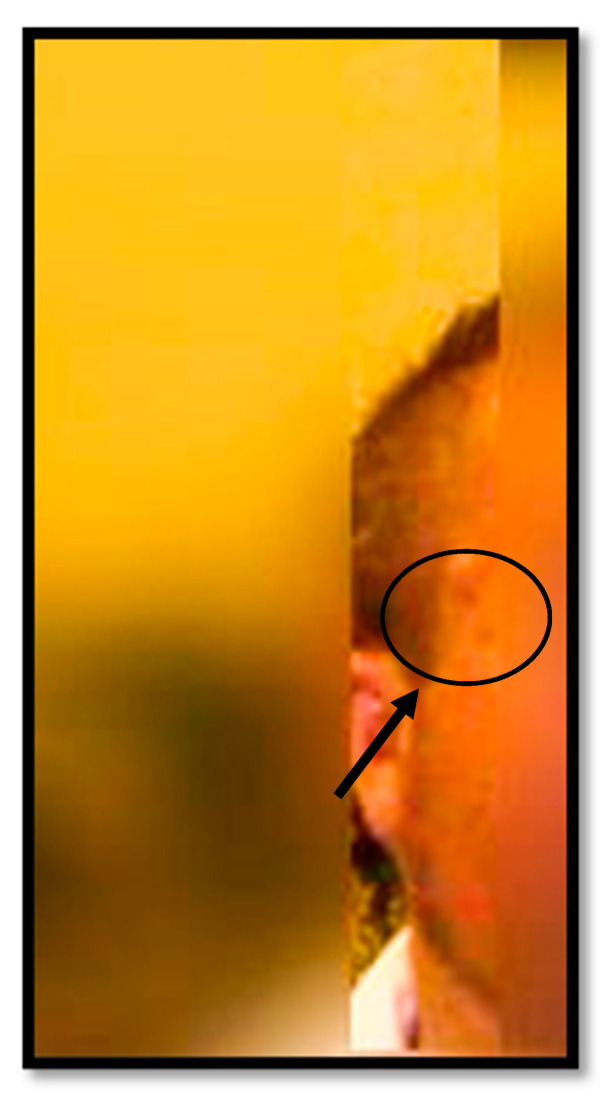
According to Pierre Bourdieu, photography is an instrument of social ritual that becomes part of everyday reality, fulfilling five aspects: “protection against time, communication with others and expression of feelings, self-realization, social prestige, distraction or escape” [1]. It is precisely the “protection against time” that means that in the medico-legal reality, the photograph can sometimes be the only piece of evidence on which to anchor an assessment [2]. The authors present a case of court litigation over a failure to diagnose melanoma, in which the complexity of evaluation (whether or not there was medical liability) lay precisely in the possibility of “proving” in court the work of the health care provider. Melanoma is a cutaneous malignancy resulting from the uncontrolled proliferation of melanocytes, neuroectodermal-derived cells. Currently, the incidence is growing steadily worldwide, and numerous studies suggest that it has even doubled in the past 10 years [3]. Cutaneous melanoma is estimated to have reached 100,000 new cases/year in the last decade-an increase of about 15 percent over the previous decade. Cutaneous melanoma is, in particular, dozens of times more frequent in individuals of European (Caucasian) stock than in other ethnic groups. In fact, the highest incidence rates are found in areas that are very sunny and inhabited by populations of northern European ethnicities with particularly fair skin [4]. Melanoma poses a challenge diagnostically because the lesions can mimic benign lesions such as nevi, freckles, and seborrheic keratoses, which are much more common in the population than melanomas [5]. Doctors, when they suspect the presence of melanoma, arrange for its removal and histological examination to ascertain its diagnosis. So much does not happen (guiltily) in the case where it is certain that the excised lesion has no malignant features. This now-established procedure is often a source of subsequent evaluative problems, as in the case presented by the authors. A frame of a picture showing subcutaneous lesion (black arrow and circle) with a roundish appearance, not pigmented and with intact skin. The authors present the case of a sebaceous cyst “confused” with a melanoma. On 7 August 2014, the patient was admitted to the Department of Neurosurgery due to the onset of headache, dizziness, walking symptoms and disorders for about 15 days. During the aforementioned hospitalization, he underwent encephalic MRI, which showed 2 oval formations in the temporo-nuchal (30 mm) and supra-ventricular (36 mm) sites, surrounded by discrete edematous halo with compression of the right ventricular system and slight leftward flaring of the midline structures; PET scan was performed with detection of neoplastic peritoneal localizations. He then underwent surgery for removal of the two brain neoformations and subsequent histological investigation with finding of “epitheliomorphic malignant neoplasm with phenotype and immunophenotype consistent with melanoma metastasis”, found to be HMB-45+, s100+, CK AE1/AE3- and negative for the BRAF gene in a test performed by Polymerase Chain Reaction (PCR) technique. Therefore, during the hospitalization, dermatologic and ophthalmologic consultations were performed at which no suspicious/primitive neoplastic lesions emerged. During dermatologic consultation, no pigmentary lesions suspicious for melanoma were revealed, but the patient reported that he had removed a nodular lesion of no known significance in the right temporal region. However, there was no diagnostic report of that surgery. The patient then died the following year as a result of complications given by metastasis of melanoma to neoplastic meningiosis of the spine, with a consensual hemorrhagic component. The patient’s family filed a claim against the dermatology physician who had removed a subcutaneous lesion in the right temporal region a few years earlier by failing to submit it for histological examination, thus complaining of a plausible failure to diagnose melanoma; due to complications, the patient died. During legal operations, the dermatology physician claimed to have removed a subcutaneous lesion, about 1 year before the 2018 hospitalization, by outpatient surgery. The same physician claimed that the lesion had features attributable to a sebaceous cyst and, therefore, in his experience, did not require further investigation by histologic examination. Of the above, he did not produce any report. The patient’s family members, who had filed for compensation, exhibited some photographs in which the lesion in dispute could be observed. In the photographs taken prior to the dermatologist’s intervention, it was evident, in the right temporal region, below the lateral end of the eyebrow arch (used as a repere), the presence of two lesions: a subcutaneous lesion with a roundish appearance, not pigmented and with intact skin (Figure 1) and, inferior to this, a pigmented lesion with a very dark coloring (Figure 2).

**Figure 2 diagnostics-13-01033-f002:**
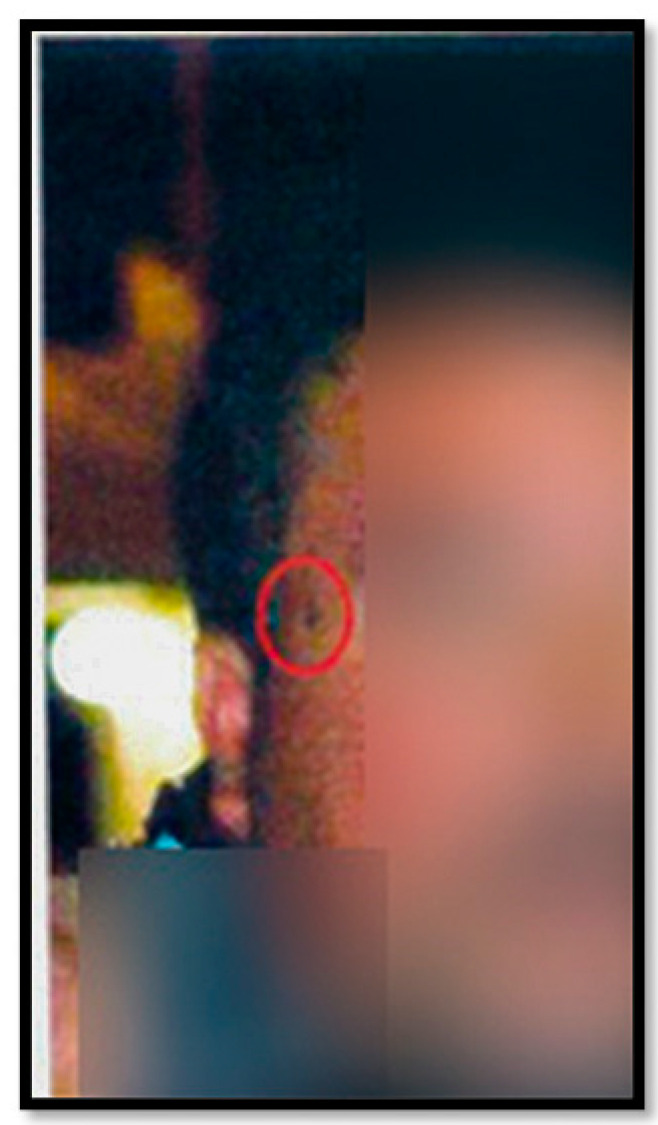
A frame of a picture showing a pigmented lesion with a very dark coloring and irregular borders (red circle).

**Figure 3 diagnostics-13-01033-f003:**
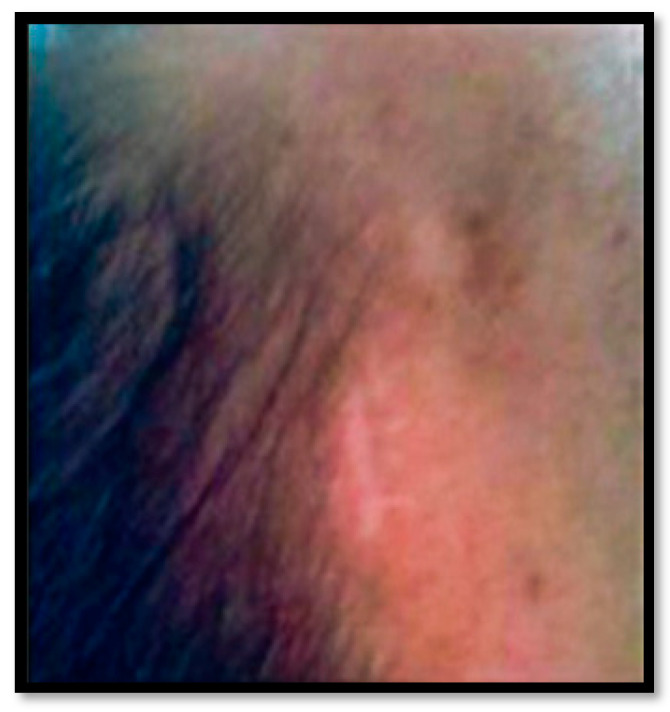
A frame of a picture showing a scarring outcome of a few cm, inveterate, normotrophic, and hypopigmented compared with the surrounding skin. In other frames following Dr. T.A.’s surgery, however, it was possible to observe in the right temporal region, posterior to the eyebrow arch (always used as a repere), a scarring outcome of a few cm, inveterate, normotrophic, hypopigmented compared with the surrounding skin. However, there was no evidence of either the nodular lesion or the pigmented lesion. It was thus possible to infer that the dermatologist had not removed the pigmented lesion in question, but the subcutaneous lesion located superior to it, namely the sebaceous cyst. If in fact, he had removed a melanoma or metastasis of melanoma, having made a small incision, such that a neoplastic lesion (extending into the deeper layers of the skin) could not be completely removed, the melanoma would have reformed in the same location in the immediate following months. Furthermore, since the cyst does not go into regression, it is reasonable to assume that if he had removed the pigmented lesion, there would have been photographic or documentary evidence (in the dermatologic consultation performed during the hospitalization) of a cystic lesion. It is highly likely that the pigmented lesion in question has gone through the phenomenon of regression. In fact, in between 4 and 12 percent of cases, the site of melanoma onset is not recognizable, and the diagnosis is made by the finding of a metastasis in the absence of a clinically evident primary melanoma. This phenomenon can be explained by assuming the presence of a spontaneously regressed cutaneous melanoma. A previous lesion that spontaneously disappeared years before the appearance of metastasis is, in fact, reported in 10–20% of cases in the literature. Histopathologic regression consists, in fact, of partial (segmental) or complete replacement/obliteration of tumor cells associated with mononuclear inflammatory infiltrate, melanophages and/or dermal neovascularization and fibrosis and, therefore, results in the disappearance of the lesion. Photographic images provided by family members together with knowledge regarding the diagnosis of melanoma allowed the case to be reconstructed and the conclusion to be reached that the physician had acted according to science and conscience by removing a benign lesion. The case presented is an example of medico-legal litigation in dermatology [6]. Claims for compensation for failure to diagnose melanoma are very frequent, both because of the inherent characteristics of the neoplasm and because of the physician’s failure to provide detailed documentation. Claims related to cases of omitted diagnosis of melanoma are, in general, due to the alleged failure to diagnose this neoplasm in a timely manner [7,8]. The most common scenarios that result in the misdiagnosis of melanoma are (1) misdiagnosis of nodular melanoma by a dermatologist; (2) misdiagnosis of nodular melanoma by the anatomo-pathologist; (3) incomplete biopsy; (4) melanoma misdiagnosed as “dysplastic nevus with edge involvement”; (5) melanoma misdiagnosed as Spitz’s nevus; (6) unrecognized desmoplastic melanoma and (7) metastatic disease without known primary disease [9]. It will be useful to go over some features of melanoma and its diagnosis in extreme summary. There are different subgroups depending on certain clinical and histopathological features; for the histological classification of melanoma, reference is made to the 2018 WHO classification, which includes, among others, some of the most frequent types of melanoma, including superficial spreading melanoma (SMM), nodular melanoma (NM), lentigo maligna (LM), and acral-lentiginous melanoma (ALM). The clinical diagnosis of melanoma is based on clinical-morphological aspects and dermatoscopic examination [10]. The clinical parameters to be considered are the ABCDE criteria: asymmetry (A), irregularity in borders (B) and color (C), dimensional increase (D), and evolutions within the lesion (E). The ‘dermatoscopic examination can improve diagnostic accuracy by 15–20% over simple clinical observation [11]. Melanoma arises at or contiguous with a preexisting congenital acquired melanocytic nevus, in up to half of the cases [12]. In the remaining cases, it arises in a skin area not the site of melanocytic nevi. The major insidiousness is the occurrence of melanomas at sites other than the skin, such as at the eye or mucosal level (oral cavity, nasal mucosa, vulva and vagina, anus and anorectal canal, penis), or at the nail level. Rare, however, are primary melanomas starting in internal organs [13]. However, the diagnostic confirmation of melanoma is histologic, so in cases of suspected melanoma, a histologic examination is mandated, which allows for the definition of the histotype, the growth phase of the neoplasm, and the evaluation of some parameters that are of fundamental importance with regard to prognosis (Breslow’s thickness, number of mitoses/mmq, ulceration, the possible presence of regression, lymphovascular invasion, and neurotropism) [14,15]. Ranging from 15% to 35%, patients with cutaneous melanoma without metastasis at the time of first diagnosis subsequently develop disease progression. Metastasis can occur by contiguity, by the lymphatic route (to regional lymph nodes or in transit) or by the bloodstream. Spread generally occurs through three modes: direct extension consisting of horizontal and vertical growth stages; lymphatic spread by embolization through intradermal vessels with skin localizations near or distant from the primary site and hematic spread [16]. In 4 to 12 percent of cases, the site of melanoma onset is not recognizable, and the diagnosis is made by the finding of a metastasis, usually a lymph node, in the absence of a clinically evident primary melanoma [8]. This phenomenon, once the presence of mucosal or ocular melanoma is ruled out, can be explained by assuming the presence of a spontaneously regressed cutaneous melanoma or, more rarely, primary lymph node-onset melanoma. A previous lesion that disappeared spontaneously or cauterized or excised several years before the appearance of metastasis is reported in 10–20% of cases [17]. These are the main reasons why dermatology physicians are frequently victims of complaints. Very often, in fact, the diagnosis of melanoma conceals a skin lesion excised a few months earlier, and so the patient or his family members decide to file a complaint against the physician. Very often, the lack of physician protection leads to a shift from clinical prevention activity to litigation prevention. This problem plaguing medicine is not just an Italian problem, but a truly global issue. Even in the United States of America (USA), there have been numerous studies that have addressed the issue of litigation in dermatology. Daniel D. Lydiatt, using the analysis of jury verdicts, conducted an analysis of lawsuits in cutaneous melanoma cases, stating that a failure to diagnose was alleged in 54% of the cases and that, of these, only 48% of the judgments saw a concrete omission of the diagnostic test [7]. This must inevitably give pause to the amount of litigation initiated on unfounded grounds. Unfortunately, so many results in serious harm to the physician, with an inevitable drift of medical work from patient care to personal protection in the view of possible judgment. In patients who present with a diagnosis of metastasis from melanoma in the absence of a primary lesion, following the removal of a ‘benign’ lesion in whom the site of the primary melanoma cannot be located, it may be assumed that the excised or destroyed ‘benign’ lesion was in fact a melanoma [16,18]. One means of avoiding this scenario is to perform a histological examination for all excised lesions, even those that are apparently benign [8]. This will inevitably lead to increased costs and longer waiting lists for histologic diagnosis, with the downside that in those truly affected by neoplastic disease there will be delayed initiation of therapy. However, is this really the direction to go? Is it right to put judicial protection first rather than patient care? In this scenario of “malpractice hunting,” the physician must be better protected and must, therefore, have the tools to defend himself, considering that the diagnosis of melanoma, as we have seen in our case, must be timely and the survival rate depends precisely on the celerity of the diagnosis [19,20,21,22]. Actually, it would suffice to comply with a protocol to be followed in the case of skin lesions that need removal. The physician in charge must create a file for each patient in which, in addition to noting the medical history, he or she must include a detailed description of the lesion to be removed and, with the patient’s consent, attach photographic documentation of it. For each lesion removed, it will be helpful to describe: the type, shape, color, size, and anatomic location, indicate whether it is a lesion detected in relation to the surrounding skin, and approximate date of the appearance of the lesion. It will, in addition, be useful to take photographs in which the location of the lesion can be pinpointed, and its macroscopic features identified [17]. Even more useful would be to also attach any photographs taken with the use of a dermatoscope. Such a simple stratagem would make it possible not to overshadow patient care, while also ensuring elements of high evidentiary power in court. The possibility of attesting to what has been conducted with documentary elements (photos and description), guarantees protection for the physician who will, therefore, be able to always operate in the best interest of the patient, but in full self-defense in the possible judicial context. If in the case presented, the dermatology physician had operated with these indications, the doubt would probably not have arisen that he had removed a neoplastic lesion. The case was, in fact, able to be resolved occasionally with the help of amateur photographs of the deceased. So much demonstrates the importance of iconographic documentation, especially when excising benign lesions. Small shrewdness could, therefore, secure the physician, and avoid numerous claims, attested, however, histological examination is mandated always since clinical and dermoscopic features are never diagnostic, but only indicative, sometimes with excellent reliability, but in other cases, precisely because of the intrinsic characteristics of some skin lesions, absolutely unreliable.

**Figure 4 diagnostics-13-01033-f004:**
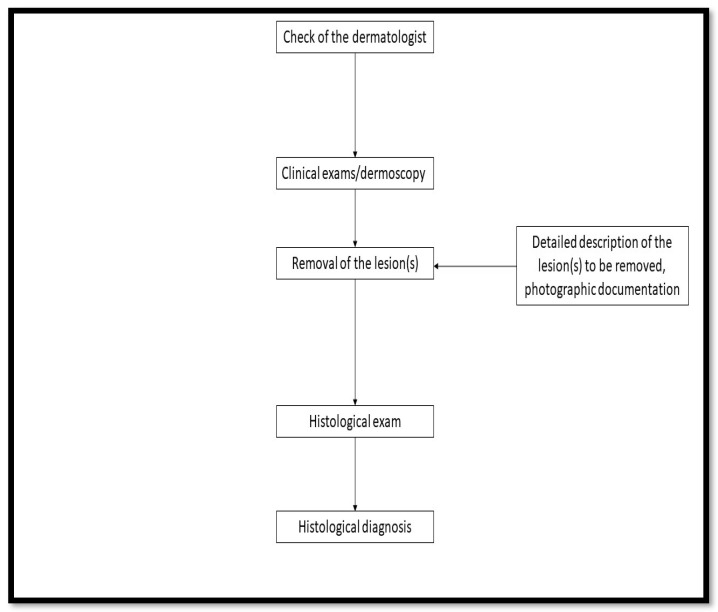
The authors, therefore, propose a protocol that is as clinically useful as it is helpful in avoiding medical-legal issues (Figure 4). Flow chart of the proposed protocol. It is important to document and keep track of all medical consultations, but this should not lead to a wild bureaucratization of care: the health and care of the patient should always come first.

## Data Availability

Not applicable.

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
