# Peer review of "Photography as the Sole Means of Proof: Medical Liability in Dermatology"

_diagnostics, 2023, doi:10.3390/diagnostics13061033_

Round 1

Reviewer 1 Report

The manuscript by Marrone et al. provides a description of a paradigmatic case resulting in medico-legal litigation in dermatology, following a resection that did not undergo histological examination. Briefly, a patient presented with what appeared to be a sebaceous cyst which was removed, but not submitted to histological examination. A year later, the patient returned with two brain metastases consistent with melanoma, which one year later resulted in the patient’s death. A claim was later filed against the physician who had removed the lesion but not requested histologic examination – an occurrence which the authors point out is both frequent and problematic for physicians and patients alike. Hence, the authors propose the careful documentation of all lesions removed including, critically, photographic evidence, which was found to be sufficient to rule out medical negligence.

The manuscript is well written, makes a fine job of raising physician’s attention to the subject, and provides a simple yet empirically proven solution to the issue. While this reviewer understands the authors’ reluctance to give an example of what would be, in their eyes, an appropriate documentation protocol, this would be critical for completing the picture. Could the authors provide – in figure or table format (which would further catch the reader’s attention) – a succinct, generalizable protocol and/or example?

Minor points:

-          Is “for” (line 49) the word the authors meant to use?

-          Is reference 6 (line 124) the reference the authors meant to produce?

-          Line 178: Consider the following alternative “Even in the United States of America”, or “Even in the Americas/American continent, in the United States”

-          Line 185: “result” as many is plural

-          Line 208: typo in “indicate”

Author Response

Dear Reviewer n'1,

thank you very much.

We modified the manuscript in according to your tips.

Thanks a lot

Reviewer 2 Report

The quality of the pictures is very low. Please try to insert more qualitative pictures.

In the discussion, the authors have to include more recently published articles.

The article has no conclusion.

Author Response

Dear Reviewer n'2,

thank you very much. We improved the quality of our pictures, even if the "goal" of our paper is to demonstrate that only "amatorial" elements such these pictures allowed to help the dermatologist. Furthermore, we added some update references in discussion and, finally, I added a conclusion section.